# Enhancing Retrieval-Augmented Large Language Models with Iterative Retrieval-Generation Synergy

**Zhihong Shao[1], Yeyun Gong[2], yelong shen[3], Minlie Huang[1]\*, Nan Duan[2], Weizhu Chen[3]**

[1] The CoAI Group, DCST, Institute for Artificial Intelligence,
[1] State Key Lab of Intelligent Technology and Systems,
[1] Beijing National Research Center for Information Science and Technology,
[1] Tsinghua University, Beijing 100084, China
[2] Microsoft Research Asia [3] Microsoft Azure AI
szh19@mails.tsinghua.edu.cn aihuang@tsinghua.edu.cn

## Abstract

Retrieval-augmented generation has raise extensive attention as it is promising to address the limitations of large language models including outdated knowledge and hallucinations. However, retrievers struggle to capture relevance, especially for queries with complex information needs. Recent work has proposed to improve relevance modeling by having large language models actively involved in retrieval, i.e., to guide retrieval with generation. In this paper, we show that strong performance can be achieved by a method we call ITER-RETGEN, which synergizes retrieval and generation in an iterative manner: a model's response to a task input shows what might be needed to finish the task, and thus can serve as an informative context for retrieving more relevant knowledge which in turn helps generate a better response in another iteration. Compared with recent work which interleaves retrieval with generation when completing a single output, ITER-RETGEN processes all retrieved knowledge as a whole and largely preserves the flexibility in generation without structural constraints. We evaluate ITER-RETGEN on multi-hop question answering, fact verification, and commonsense reasoning, and show that it can flexibly leverage parametric knowledge and non-parametric knowledge, and is superior to or competitive with state-of-the-art retrieval-augmented baselines while causing fewer overheads of retrieval and generation. We can further improve performance via generation-augmented retrieval adaptation.

## 1 Introduction

Generative Large Language Models (LLMs) have powered numerous applications, with well-perceived utility. Despite being powerful, LLMs lack knowledge that is under-represented in their training data, and are prone to hallucinations, especially in open-domain settings (OpenAI, 2023).

Retrieval-augmented LLMs, therefore, have raised widespread attention as LLM outputs can be potentially grounded on external knowledge.

Previous retrieval-augmented LMs (Izacard et al., 2022b; Shi et al., 2023) typically adopted one-time retrieval, i.e., to retrieve knowledge using only the task input (e.g., a user question for open-domain question answering). One-time retrieval should suffice to fulfill the information needs if they are clearly stated in the original input, which is applicable to factoid question answering (Kwiatkowski et al., 2019) and single-hop fact verification (Thorne et al., 2018), but not to tasks with complex information needs, e.g., multi-hop reasoning (Yang et al., 2018) and long-form question answering (Fan et al., 2019).

To fulfill complex information needs, recent work proposes to gather required knowledge multiple times throughout the generation process, using partial generation (Trivedi et al., 2022a; Press et al., 2022)) or forward-looking sentence(s) (Jiang et al., 2023) as search queries. However, such structured workflows of interleaving retrieval with generation have the following limitations: (1) as intermediate generation is conditioned on knowledge retrieved before, with no awareness of knowledge retrieved afterwards, they fail to process all retrieved knowledge as a whole during the generation process; (2) they require multi-round retrieval to gather a comprehensive set of knowledge, and may frequently change the prompts by updating newly retrieved knowledge, thus increasing the overheads of both retrieval and generation.

In this paper, we find it simple but effective to enhance retrieval-augmented LLMs through iterative retrieval-generation synergy (ITER-RETGEN, Fig 1). ITER-RETGEN iterates *retrieval-augmented generation* and *generation-augmented retrieval*: Retrieval-augmented generation outputs a response to a task input based on all retrieved knowledge (initially using the task input as the query). This

---

*Corresponding author: Minlie Huang.

output shows what might be needed to fulfill the task, and thus can serve as an informative context to retrieve more relevant knowledge, i.e., generation-augmented retrieval. The newly retrieved knowledge can benefit another iteration of retrieval-augmented generation. We can also leverage model generations to adapt retrieval, by distilling knowledge from a re-ranker with access to model generations to a dense retriever with access to task inputs only, which may be beneficial in scenarios where user inputs can be easily collected, but relevant knowledge or desirable outputs are not annotated.

We evaluate our method on three tasks, including multi-hop question answering, fact verification, and commonsense reasoning. Our method prompts an LLM to produce a chain of reasoning steps followed by the final answer under a few-shot setting. For in-context demonstrations, we focus on problem-solving and follow Wei et al. (2022) to annotate chains of thoughts, without explicitly considering how generation-augmented retrieval might be affected, which makes it conceptually simple and easy to implement. Our method achieves up to 8.6% absolute gains over previous state-of-the-art retrieval-augmented methods on four out of six datasets while being competitive on the remaining two. According to our experiments, generation generally benefits from more iterations, with two iterations giving the most performance gains. One may customize the performance-cost tradeoffs by choosing an appropriate number of iterations. We can further improve performance and also reduce iterations via the aforementioned generation-augmented retrieval adaptation.

We summarize our findings as follows:

- Automatic metrics such as exact match can significantly underestimate the performance of LLMs in question answering tasks. Moreover, improvements in exact match do not always reflect improvements in generations. Evaluation using LLMs may be more reliable.

- ITER-RETGEN is superior to or competitive with state-of-the-art retrieval-augmented methods, while being simpler and causing fewer overheads of retrieval and generation. With generation-augmented retrieval adaptation, we can further improve performance and also reduce overheads (by reducing iterations).

- It is desirable for an LLM to leverage both

parametric knowledge and non-parametric knowledge effectively. ITER-RETGEN consistently outperforms Self-Ask on question answering tasks, regardless of whether in-context non-parametric knowledge mentions the answers or not.

## 2 Related Work

In recent months, there has been a surge in LLM-powered applications, such as ChatGPT, Bing Chat, and CoPilot (Chen et al., 2021). While showing an unprecedented level of performance, LLMs are subject to the following limitations: (1) Due to a high demand for compute and data, it remains an open research question to continually update LLMs both efficiently and effectively (Scialom et al., 2022); (2) LLMs also tend to hallucinate (OpenAI, 2023), i.e., generating plausible but non-factual texts. To alleviate these issues, there is a growing trend of augmenting LLMs with tools (Mialon et al., 2023; Gou et al., 2023), e.g., a code interpreter (Gao et al., 2022b; Shao et al., 2023) or a search engine (Nakano et al., 2021), in an attempt to offload subtasks to more qualified experts, or to enrich the input context for LLMs by providing more relevant information.

Retrieval augmentation is a mainstream direction to connect LLMs to the external world. Previous retrieval-augmented LMs (Izacard and Grave, 2021; Shao and Huang, 2022) typically receive retrieved knowledge in a passive way: knowledge is retrieved based on the task inputs without LMs' intervention. As it is difficult for a retriever to capture relevance, especially in the zero-shot setting, recent work shows a shift towards having LLMs actively involved in retrieval to improve relevance modeling, e.g., to provide a specific context for retrieval with model generations (e.g., generated search queries (Nakano et al., 2021; Press et al., 2022; Yao et al., 2022), partial generation (Trivedi et al., 2022a), or forward-looking sentences (Jiang et al., 2023)). Khattab et al. (2022) proposed a DSP programming framework that supports various retrieval-augmented methods.

Recent work interleaves retrieval with generation when completing a single output. Such a structured workflow may reduce the flexibility in generation (Yao et al., 2022). ITER-RETGEN avoids interrupting generation with retrieval, but iterates retrieval and generation, i.e., to leverage the complete generation from the previous iteration to retrieve more

**Question:** $q$

**What is the height of the player who won the 2015 AFL Rising Star award?**

**Retrieval:** $q \rightarrow \mathcal{D}_q$

**Title: 2015 AFL Rising Star Context: The NAB AFL Rising Star award is given annually to a stand out young player in the Australian Football League. The award was won by Jesse Hogan of Melbourne**

**Retrieval-Augmented Generation:** $\mathcal{D}_q \parallel q \rightarrow y_1$

**The 2015 AFL Rising Star award was won by Jesse Hogan of Melbourne. Jesse Hogan is a professional Australian rules footballer. He is 198 cm tall. So the answer is 198 cm**

**Iteration 1**

**Question:** $q$

**What is the height of the player who won the 2015 AFL Rising Star award?**

**Generation-Augmented Retrieval:** $y_1 \parallel q \rightarrow \mathcal{D}_{y_1 \parallel q}$

**Title: Jesse Hogan Context: Jesse Hogan … playing for the Melbourne Football Club. A key forward, Hogan is 1.95 m tall … made his AFL debut in the 2015 season and won the Ron Evans Medal as the AFL Rising Star**

**Retrieval-Augmented Generation:** $\mathcal{D}_{y_1 \parallel q} \parallel q \rightarrow y_2$

**The 2015 AFL Rising Star award was won by Jesse Hogan of Melbourne. Jesse Hogan is 1.95 m tall. So the answer is 1.95 m**

**Iteration 2**

Figure 1: ITER-RETGEN iterates retrieval and generation. In each iteration, ITER-RETGEN leverages the model output from the previous iteration as a specific context to help retrieve more relevant knowledge, which may help improve model generation (e.g., correcting the height of Hesse Hogan in this figure). We only show two iterations in this figure for brevity. Solid arrows connect queries to the retrieved knowledge, and dashed arrows denote retrieval-augmented generation.

relevant information which helps improve generation in the next iteration. ITER-RETGEN also has the advantage of processing all retrieved knowledge as a whole during the generation process, and is conceptually simpler and easier-to-implement, while being empirically strong in multi-hop question answering, fact verification, and commonsense reasoning.

A closely related work called GAR (Mao et al., 2021) augments queries with generated background information. HyDE (Gao et al., 2022a) also shares a similar spirit, but focuses on zero-shot information retrieval, and proposes to first prompt an LLM to produce "hypothetical" paragraphs that cover the information needed to answer a given question, and then use the generated paragraphs to retrieve the real ones. RepoCoder (Zhang et al., 2023) focuses on repository-level code completion, and proposes a 2-iteration retrieval-generation paradigm where the second iteration leverages the intermediate code completion for retrieval. By contrast, we propose to synergize retrieval and generation with ITER-RETGEN on various natural language tasks, and explore how we can further adapt retrieval with model generations.

## 3 Iterative Retrieval-Generation Synergy

### 3.1 Overview

Given a question $q$ and a retrieval corpus $\mathcal{D} = \{d\}$ where $d$ is a paragraph, ITER-RETGEN repeats retrieval-generation for $T$ iterations; in iteration $t$, we (1) leverage the generation $y_{t-1}$ from the previous iteration, concatenated with $q$, to retrieve

top-$k$ paragraphs, and then (2) prompt an LLM $\mathcal{M}$ to produce an output $y_t$, with both the retrieved paragraphs (denoted as $\mathcal{D}_{y_{t-1} \parallel q}$) and $q$ integrated into the prompt. Therefore, each iteration can be formulated as follows:

$$y_t = \mathcal{M}(y_t | \mathtt{prompt}(\mathcal{D}_{y_{t-1} \parallel q}, q)), \ \forall 1 \le t \le T \quad (1)$$

The last output $y_T$ will be produced as the final response.

### 3.2 Generation-Augmented Retrieval

There are many natural language tasks with complex information needs. For example, in open-domain multi-hop question answering, specific information needs may manifest themselves only after correctly answering some prerequisite sub-questions. In other words, there may exist semantic gaps between the original question $q$ and its supporting knowledge, which can not be effectively addressed by a retriever with a representation bottleneck. In the first iteration, we can retrieve knowledge with only the question $q$. In later iterations, the LLM output from the previous iteration, though having no guarantee of correctness, shows what might be needed to answer the question, and thus can be leveraged to bridge the semantic gaps; with improved retrieval, an LLM can potentially produce a better output.

### 3.3 Retrieval-Augmented Generation

In each iteration, we generate an output using Chain-of-Thought prompting except that we also prepend retrieved knowledge to the question $q$. Though there may exist more advanced prompting

variants, e.g., incorporating previous generations into the prompt to enable direct refinements, we leave the explorations for future work, and focus on investigating the synergy between retrieval and generation in a straightforward manner.

### 3.4 Generation-Augmented Retrieval Adaptation

Model generations not only provide specific contexts for retrieval, but can also be leveraged to optimize the retriever, so that information needs in a question can be better captured by the retriever.

**Dense Retriever** We adopted dense retrieval in our experiments. Given a dense retriever parametrized by $\theta = \{\theta_q, \theta_d\}$ where $\theta_q$ and $\theta_d$ denote parameters of the query encoder and the paragraph encoder, respectively, the similarity score between a query and a paragraph is calculated as the inner product of their encoded vectors:

$$s_\theta(q, d) = \langle \mathbf{E}(q; \theta_q), \mathbf{E}(d; \theta_d) \rangle \quad (2)$$

**Re-ranker** A re-ranker, parametrized by $\phi$, outputs the probability of a paragraph being relevant to a query; we denote the probability as $s_\phi(q, d)$.

**Distillation** A re-ranker is typically better at capturing relevance between a query and a paragraph than a retriever. Therefore, we distill knowledge from a re-ranker to a retriever. To help the retriever better address the semantic gaps between a question and its supporting knowledge, we allow access to $y_1$ for the re-ranker (where $y_1$ is the LLM output from the first iteration). We optimize only the query encoder of the retriever using the following training objective:

$$\theta_q^* = arg \min_{\theta_q} \text{KL}(P_\phi(\cdot|y_1, q), P_\theta(\cdot|q))$$

$$P_\phi(d|y_1, q) = \frac{\exp(s_\phi(y_1||q, d)/\tau)}{\sum_{d' \in \mathcal{D}_{y_1||q}} \exp(s_\phi(y_1||q, d')/\tau)} \quad (3)$$

$$P_\theta(d|q) = \frac{\exp(s_\theta(q, d)/\tau)}{\sum_{d' \in \mathcal{D}_{y_1||q}} \exp(s_\theta(q, d')/\tau)}$$

where $\text{KL}(\cdot, \cdot)$ denotes the KL divergence between two probabilistic distributions.

## 4 Experiments

### 4.1 Datasets

We experimented on six datasets of three reasoning tasks: (1) **Multi-hop question answering**, including HotPotQA (Yang et al., 2018), 2WikiMultiHopQA (Ho et al., 2020), MuSiQue (Trivedi et al., 2022b), and Bamboogle (Press et al., 2022). On MuSiQue, we followed Press et al. (2022) to use only 2-hop questions; (2) **Fact Verification**, including Feverous (Aly et al., 2021); (3) **Commonsense reasoning**, including StrategyQA (Geva et al., 2021). Examples are presented in Table 1.

We used the October 2017 (Yang et al., 2018) and the December 2018 (Karpukhin et al., 2020) Wikipedia dump as the retrieval corpus for HotPotQA and 2WikiMultiHopQA, respectively, and used the December 2021 Wikipedia dump (Izacard et al., 2022b) for the other datasets.

### 4.2 Evaluation Settings

We conducted evaluations on all 125 questions from Bamboogle, the first 500 questions from the train set of StrategyQA, and the first 500 questions from the development sets of the other datasets. All methods are evaluated under the 3-shot setting, sharing the same questions in demonstrations.

Evaluation metrics are exact match (EM) and F1 for multi-hop question answering datasets, and accuracy for both fact verification and commonsense reasoning datasets. For more robust evaluation, we also evaluate the correctness of model outputs using `text-davinci-003`, the resulting metric denoted as Acc[†]. The prompt used for evaluation is as follows, where {question}, {model output}, and {answer} are placeholders.

---

**Prompt for Evaluating the Correctness of a Model Output**

In the following task, you are given a Question, a model Prediction for the Question, and a Ground-truth Answer to the Question. You should decide whether the model Prediction implies the Ground-truth Answer.

Question
{question}

Prediction
{model output}

Ground-truth Answer
{answer}

Does the Prediction imply the Ground-truth Answer? Output Yes or No:

| Datasets | Example |
|---|---|
| HotPotQA | What is the name of this American musician, singer, actor, comedian, and songwriter, who worked with Modern Records and born in December 5, 1932? |
| 2WikiMultiHopQA | Which film came out first, Blind Shaft or The Mask Of Fu Manchu? |
| MuSiQue | In which year did the publisher of In Cold Blood form? |
| Bamboogle | When did the first prime minister of the Russian Empire come into office? |
| Feverous | Is it true that Based on the same platform as the Chevrolet Sail, the Baojun 310 was launched on 2017 Beijing Auto Show where the price ranges from 36.800 yuan to 60.800 yuan? |
| StrategyQA | Is it common to see frost during some college commencements? |

Table 1: Example questions from six datasets.

## 4.3 Baselines

**Direct Prompting** (Brown et al., 2020) prompts an LLM to directly generate the final answer without an explanation. When augmenting Direct prompting with retrieval, we used the question to retrieve knowledge which will be placed before the question in the prompt.

**CoT Prompting** (Wei et al., 2022) prompts an LLM to generate natural language reasoning steps followed by the final answer.

**ReAct** (Yao et al., 2022) interleaves reasoning, action, and observation steps, until reaching the action of finalizing an answer. An action can be either generating a query to search for information or finalizing an answer. An observation is the concatenation of retrieved paragraphs.

**Self-Ask** (Press et al., 2022) interleaves (i) follow-up question generation, (ii) retrieval using the follow-up, and (iii) answering the follow-up conditioned on the retrieved knowledge, until no more follow-up questions are generated and the LLM gives an answer to the original question. We followed (Yoran et al., 2023) to prepend newly retrieved paragraphs to the original question. On our evaluated tasks, Self-Ask is conceptually similar to ReAct, with the main difference being that Self-Ask accumulates retrieved knowledge before the original question in the prompt, while ReAct places retrieved knowledge right after its query. Self-Ask and IRCoT (Trivedi et al., 2022a) also share the spirit of synergizing reasoning and retrieval.

**DSP** (Khattab et al., 2022) comprises a multi-hop retrieval stage and an answer prediction stage. For each hop within the retrieval stage, the model is prompted to generate search queries and to sum-marize retrieve knowledge for subsequent use. In the prediction stage, DSP generates the answer using CoT based on the summarized knowledge and retrieved documents.

## 4.4 Implementation Details

We used `text-davinci-003` version of Instruct-GPT (Ouyang et al., 2022) as the backend LLM. We also present experiments using the open-source Llama-2 models (Touvron et al., 2023) in Appendix A. All experiments used greedy decoding. Contriever-MSMARCO (Izacard et al., 2022a) was used for retrieval. We retrieved top-5 paragraphs for each query. We allowed at most 5 interactions with retrieval for ReAct and Self-Ask. We adapted the implementation of DSP [1] to use the same generation model and retrieval systems as the other methods.

Note that the first iteration of ITER-RETGEN is CoT prompting with retrieval augmentation. Therefore, ITER-RETGEN and CoT prompting share the same annotated in-context demonstrations. All prompts are presented in the Appendix.

## 4.5 Main Results

As shown by Table 2, ITER-RETGEN ($T \geq 2$) achieve significantly higher $Acc^{\dagger}$ than retrieval-augmented baselines on HotPotQA, 2WikiMulti-HopQA, Bamboogle, and StrategyQA, while being competitive with the best method (i.e., Self-Ask) on MuSiQue and Feverous.

When increasing the number of iterations for ITER-RETGEN, performance generally improves, with the second iteration giving the greatest boost.

---

[1] https://github.com/stanfordnlp/dspy/issues/85

| Method | HotPotQA | | | 2WikiMultiHopQA | | | MuSiQue | | | Bamboogle | | | Feverous | | StrategyQA | |
|---|---|---|---|---|---|---|---|---|---|---|---|---|---|---|---|---|
| | EM | F1 | Acc$^\dagger$ | EM | F1 | Acc$^\dagger$ | EM | F1 | Acc$^\dagger$ | EM | F1 | Acc$^\dagger$ | Acc | Acc$^\dagger$ | Acc | Acc$^\dagger$ |
| | | | | | | | Without Retrieval | | | | | | | | | |
| Direct | 21.9 | 36.8 | 44.8 | 21.3 | 29.2 | 33.9 | 7.0 | 18.7 | 15.8 | 11.2 | 24.4 | 28.0 | 60.1 | 60.1 | 66.5 | 66.7 |
| CoT | 30.0 | 44.1 | 50.0 | 30.0 | 39.6 | 44.0 | 19.4 | 30.9 | 28.6 | **43.2** | **51.1** | 60.0 | 59.8 | 59.8 | 71.0 | 71.0 |
| | | | | | | | With Retrieval | | | | | | | | | |
| Direct | 31.6 | 44.7 | 53.3 | 27.3 | 35.4 | 43.6 | 13.9 | 28.2 | 26.5 | 17.6 | 31.8 | 43.2 | 69.8 | 69.8 | 65.6 | 65.6 |
| ReAct | 24.9 | 44.7 | 61.1 | 28.0 | 38.5 | 45.9 | 23.4 | 37.0 | 37.9 | 21.8 | 31.0 | 40.3 | 66.4 | 66.4 | 66.9 | 66.9 |
| Self-Ask | 36.8 | 55.2 | 64.8 | **37.3** | **48.8** | 55.9 | 27.6 | 41.5 | **42.9** | 31.5 | 41.2 | 54.8 | 70.7 | 70.7 | 70.2 | 70.2 |
| DSP | 43.8 | 55.0 | 60.8 | - | - | - | - | - | - | - | - | - | - | - | - | - |
| ITER-RETGEN 1 | 39.2 | 53.9 | 65.5 | 33.7 | 45.2 | 55.4 | 24.2 | 38.6 | 38.1 | 36.8 | 47.7 | 57.6 | 67.0 | 67.0 | 72.0 | 72.0 |
| ITER-RETGEN 2 | 44.1 | 58.6 | 71.2 | 34.9 | 47.0 | 58.1 | 26.4 | 41.1 | 41.0 | 38.4 | 48.7 | 59.2 | 68.8 | 68.8 | 73.0 | 73.0 |
| ITER-RETGEN 3 | 45.2 | 59.9 | 71.4 | 34.8 | 47.8 | 58.3 | 25.7 | 41.4 | 40.8 | 37.6 | 47.0 | 59.2 | 69.0 | 69.0 | 72.3 | 72.3 |
| ITER-RETGEN 4 | 45.8 | **61.1** | **73.4** | 36.0 | 47.4 | 58.5 | 26.7 | 41.8 | 40.8 | 38.4 | 49.6 | 60.0 | **71.5** | **71.5** | 73.8 | 73.8 |
| ITER-RETGEN 5 | 45.2 | 60.3 | 72.8 | 35.5 | 47.5 | 58.8 | 25.7 | 40.7 | 39.6 | 39.2 | 49.7 | **60.8** | 70.3 | 70.3 | 73.2 | 73.2 |
| ITER-RETGEN 6 | **45.9** | 61.0 | 73.3 | 35.5 | 48.1 | **59.4** | 25.9 | 40.5 | 39.8 | 40.0 | 50.0 | 59.2 | 70.9 | 70.9 | 72.4 | 72.4 |
| ITER-RETGEN 7 | 45.1 | 60.4 | 72.9 | 35.5 | 47.4 | 58.4 | 26.1 | **42.0** | 41.0 | 40.0 | 50.7 | **60.8** | 70.5 | 70.5 | **74.1** | **74.1** |

Table 2: Evaluation results on multi-hop question answering, fact verification, and commonsense reasoning datasets. Acc$^\dagger$ is the accuracy of model outputs evaluated with `text-davinci-003`. For ITER-RETGEN, we evaluated LLM outputs in different iterations (up to 7 iterations). Underlined metric values are higher than those of Self-Ask.

| Method | HotPotQA | | 2WikiMultiHopQA | | MuSiQue | | Bamboogle | | Feverous | | StrategyQA | |
|---|---|---|---|---|---|---|---|---|---|---|---|---|
| | # API | # Doc | # API | # Doc | # API | # Doc | # API | # Doc | # API | # Doc | # API | # Doc |
| ReAct | 2.9 | 14.3 | 3.0 | 15.0 | 2.9 | 14.4 | 2.8 | 14.1 | 2.1 | 10.6 | 2.8 | 14.2 |
| Self-Ask | 3.2 | 16.0 | 3.2 | 15.9 | 3.0 | 14.8 | 3.0 | 14.9 | 2.3 | 11.3 | 3.0 | 15.1 |

Table 3: Average numbers of API calls to `text-davinci-003` and retrieved paragraphs for ReAct and Self-Ask. Note that ITER-RETGEN ($T = 2$) achieves significantly higher or competitive Acc$^\dagger$ with fewer API calls (i.e., 2) and fewer retrieved paragraphs (5 per iteration, 10 in total).

It is worth noting that, as shown by Table 3, ITER-RETGEN ($T = 2$) is superior to or competitive with ReAct and Self-Ask using fewer API calls to the LLM (i.e., 2) and fewer retrieved paragraphs (i.e., 5 per iteration, 10 in total). ITER-RETGEN is also conceptually simple, which is to iterate retrieval-augmented CoT, without complex processing.

We also compared ITER-RETGEN with DSP which also generates the answer using CoT based on retrieved knowledge but differs in information collection and processing. In each iteration, ITER-RETGEN retrieves knowledge based on (1) the question and (2) the previous model output which shows what may be needed to answer the question. With the number of iterations increasing, we tend to obtain a more comprehensive and relevant set of knowledge. Besides, unlike DSP, we do not summarize the retrieved documents for answer generation, and thus will not introduce summarization errors. As shown in Table 2, ITER-RETGEN outperforms

DSP significantly. We manually investigate 10 random questions where DSP fails but ITER-RETGEN provides correct answers. On 40% of them, DSP fails to retrieve documents that cover the correct answers, while on 50% of them, the summarized knowledge is misleading, e.g., for the question "What occupation do Chris Menges and Aram Avakian share?", DSP generates a wrong summary "Chris Menges and Aram Avakian are both members of the American and British Societies of Cinematographers.", while the retrieved documents mention that Aram Avakian is a film editor and director, and only Chris Menges is with the American and British Societies of Cinematographers.

**Acc$^\dagger$ is a Reliable Metric** To investigate how reliable Acc$^\dagger$ is, we focused on model outputs where EM and Acc$^\dagger$ disagree, and manually checked which metric gives more correct labels. On each of the four multi-hop question answering datasets,

| Dataset | HotPotQA | | | Feverous | | |
|---|---|---|---|---|---|---|
| Retriever | Original | Distilled w/o $y_1$ | Distilled w/ $y_1$ | Original | Distilled w/o $y_1$ | Distilled w/ $y_1$ |
| ITER-RETGEN 1 | 65.5 | 67.1 | **67.7** | 67.0 | 67.3 | **70.7** |
| ITER-RETGEN 2 | 71.2 | 75.2 | **75.7** | 68.8 | 68.1 | **69.5** |

Table 4: Effect of using LLM generation $y_1$ on optimizing a dense retriever. We evaluated ITER-RETGEN on HotPotQA and Feverous in terms of Acc$^\dagger$.

| Subset | CoT ✓ | | CoT ✗ | | w/ Answer Retrieved | | w/o Answer Retrieved | |
|---|---|---|---|---|---|---|---|---|
| Method | Self-Ask | ITER-RETGEN 2 | Self-Ask | ITER-RETGEN 2 | Self-Ask | ITER-RETGEN 2 | Self-Ask | ITER-RETGEN 2 |
| HotPotQA | 77.5 | **88.0** | 52.0 | **54.4** | 78.1 | **86.9** | 29.9 | **40.8** |
| 2WikiMultiHopQA | 68.8 | **78.2** | **46.2** | 42.0 | 73.1 | **77.2** | 30.1 | **42.3** |
| MuSiQue | **68.5** | 66.9 | **32.6** | 30.7 | 72.9 | **78.9** | 12.2 | **22.9** |
| Bamboogle | 73.0 | **77.3** | 28.0 | **32.0** | 76.2 | **82.2** | 32.8 | **46.2** |

Table 5: Comparisons between Self-Ask and ITER-RETGEN ($T = 2$) on different subsets, in terms of Acc$^\dagger$. CoT ✓ is the subset of questions which CoT answers correctly without retrieval; CoT ✗ is the complement. w/ Answer Retrieved is the subset of questions for which a method (Self-Ask or ITER-RETGEN) successfully retrieves paragraphs that mention the answers; w/o Answer Retrieved is the complement. ITER-RETGEN tends to be much better at preserving the LLM's performance on questions that can be solved using CoT without retrieval, and is consistently more accurate regardless of whether retrieved knowledge mentions the answers or not.

we randomly sampled 20 model outputs from the second iteration of ITER-RETGEN, resulting in 80 samples in total. For 98.75% of samples, EM is 0 and Acc$^\dagger$ is 1, while Acc$^\dagger$ gives the correct labels 97.5% of the time, indicating that EM severely underestimates model performance. We also carried out the same evaluation for Self-Ask, and Acc$^\dagger$ gives the correct labels 98.75% of the time when it is inconsistent with EM.

Acc$^\dagger$ offers the advantage of identifying model outputs that are semantically correct, even if their surface forms differ from the annotated answers. As an illustration, for the question "Which country Jan Baptist Van Rensselaer's father is from?", the annotated answer is Dutch, while the model prediction is Netherlands, which is correct in terms of Acc$^\dagger$ but is penalized by EM.

Notably, ITER-RETGEN ($T \geq 2$) consistently demonstrate lower EM but higher Acc$^\dagger$ than Self-Ask on 2WikiMultiHopQA, suggesting that enhancements in EM do not necessarily reflect improvements in the quality of generated answers.

| Iteration | 1 | 2 | 3 | 4 | 5 | 6 | 7 |
|---|---|---|---|---|---|---|---|
| HotPotQA | 49.5 | 66.1 | 65.7 | 66.5 | 66.7 | 66.7 | 67.1 |
| 2WikiMultiHopQA | 29.0 | 45.2 | 46.2 | 46.7 | 45.8 | 45.8 | 46.5 |
| MuSiQue | 18.6 | 32.3 | 32.3 | 33.7 | 32.7 | 33.5 | 32.9 |
| Bamboogle | 20.8 | 36.0 | 36.8 | 36.0 | 35.2 | 36.0 | 36.0 |

Table 6: Answer recall of retrieved paragraphs in different iterations for ITER-RETGEN.

**Generation Benefits Retrieval Adaptation** To investigate how LLM outputs can be leveraged for retrieval adaptation, we experimented on HotPotQA and Feverous. Specifically, on each dataset, we sampled 9,000 random questions from the train set for training, and 1,000 for validation. We applied ITER-RETGEN for one iteration, and used the model outputs $y_1$ for retrieval adaptation as in Section 3.4. We used TART (Asai et al., 2022) as the re-ranker, and distilled knowledge from TART to the dense retriever for no more than 1,000 steps. Batch size was 32 and learning rate was 1e-5. We used the retriever checkpoint with the lowest distillation loss.

As shown by Table 4, retrieval adaptation enables ITER-RETGEN to achieve significantly higher Acc$^\dagger$ with fewer iterations. We also demonstrated the benefits of using $y_1$ for adaptation by showing its improvements over a variant which only differs in that the re-ranker has no access to $y_1$; the training objective of this variant can be obtained by removing all $y_1$ notations in Eq. 3.

### 4.6 Ablation Study

#### 4.6.1 Generation Augments Retrieval

Table 6 shows the answer recall of retrieval in different iterations. The first iteration uses only the questions for retrieval and suffers from low answer recall. In the second iteration, retrieval, augmented with the LLM output from the first iteration,

| **HotPotQA** |
|---|

**Question:** The arena where the Lewiston Maineiacs played their home games can seat how many people?
**Answer:** 3,677 seated

**Iteration 1**

**Retrieved Knowledge**
(1) Title: Lewiston Maineiacs Context: The Lewiston Maineiacs were a junior ice hockey team ... The team played its home games at the Androscoggin Bank Colisée ...
(2) Title: Bangor Auditorium Context: ... a 5,948 seat multi-purpose arena located in Bangor, Maine ...
**Generation**
The Lewiston Maineiacs were a junior ice hockey team of the Quebec Major Junior Hockey League based in Lewiston, Maine. The team played its home games at the Androscoggin Bank Colisée. The Androscoggin Bank Colisée can seat up to 5,948 people. So the answer is 5,948

**Iteration 2**

**Retrieved Knowledge**
(1) Title: Lewiston Maineiacs Context: The Lewiston Maineiacs were a junior ice hockey team ... The team played its home games at the Androscoggin Bank Colisée ...
(2) Title: Androscoggin Bank Colisée Context: The Androscoggin Bank Colisée (formerly Central Maine Civic Center and Lewiston Colisee) is a 4,000 capacity (3,677 seated) multi-purpose arena ...
**Generation**
The Lewiston Maineiacs were a junior ice hockey team of the Quebec Major Junior Hockey League based in Lewiston, Maine. They played their home games at the Androscoggin Bank Colisée. The Androscoggin Bank Colisée has a capacity of 4,000 (3,677 seated). So the answer is 3,677

| **StrategyQA** |
|---|

**Question:** Can you get Raclette in YMCA headquarters city?
**Answer:** Yes

**Iteration 1**

**Retrieved Knowledge**
(1) Title: YMCA Building (Racine, Wisconsin) Context: ... Since the 50's, headquarters of the publishing house and the bookstore are located in 11 rue de la Montagne Sainte-Geneviève 75005 Paris ...
**Generation**
The YMCA headquarters is located in Paris, France. Raclette is a traditional Swiss dish, made with melted cheese served with potatoes, pickles, and other accompaniments. It is not a traditional French dish, so it is unlikely to be found in Paris. So the answer is No

**Iteration 2**

**Retrieved Knowledge**
(1) Title: Raclette Context: Raclette is a Swiss dish, also popular in France, based on heating cheese and scraping off the melted part ...
(2) Title: YMCA Building (Racine, Wisconsin) Context: ... Since the 50's, headquarters of the publishing house and the bookstore are located in 11 rue de la Montagne Sainte-Geneviève 75005 Paris ...
**Generation**
YMCA headquarters is located in Paris, France. Raclette is a dish native to parts of Switzerland, but it is also popular in France. So it is likely that Raclette can be found in Paris. So the answer is Yes

Table 7: Two examples demonstrating retrieval-generation synergy. We underline generated phrases that help retrieve relevant knowledge which successfully corrects factual errors (in red) in the second iteration. Irrelevant retrieved paragraphs are not shown in the table for brevity.

achieves significantly higher recall, indicating that LLM generations can help bridge the semantic gaps between complex questions and their supporting knowledge. However, performance quickly hits a plateau afterwards.

### 4.6.2 ITER-RETGEN Leverages Parametric and Non-Parametric Knowledge Better

Ideally, an LLM should flexibly utilize non-parametric knowledge or parametric knowledge depending on whether in-context non-parametric knowledge is relevant or not. Table 5 presents performance breakdowns on different subsets of questions for investigation. We considered the ability of CoT to answer a question correctly without re-

trieval as a proxy for assessing an LLM's capability to answer the question using its parametric knowledge. Compared with Self-Ask, ITER-RETGEN tends to be significantly better at preserving the LLM's performance on questions that the LLM can solve using CoT without retrieval, while being competitive on the complementary subset. This may be because the structural constraints from Self-Ask makes an LLM over-sensitive to the precision and comprehensiveness of follow-up question generation and answering, and Self-Ask is also incapable of processing all retrieved knowledge as a whole, thus reducing the LLM's flexibility in solving a question. Moreover, ITER-RETGEN consistently outperforms Self-Ask by a large margin, regardless

of whether the in-context non-parametric knowledge mentions the answers or not. This indicates that when the in-context non-parametric knowledge is irrelevant or incomplete, ITER-RETGEN exploits parametric knowledge better than Self-Ask.

### 4.7 Error Analysis

On HotPotQA, we manually analyzed 20 random cases where ITER-RETGEN ($T = 2$) fails. 25% of predictions are false negatives. On 10% of cases, ITER-RETGEN retrieves all necessary information but fails to perform correct reasoning. The remaining 65% of error cases are related with retrieval, on 76.9% of which, retrieval is misled by completely wrong reasoning from the first iteration, while on the other cases, reasoning in the first iteration is partially correct, but the retriever fails to retrieve the missing pieces in the second iteration. We also observed that, in the first iteration, reasoning can be negatively affected by noisy and possibly distractive knowledge retrieved using only the questions as the queries.

## 5 Case Study

Table 7 demonstrates retrieval-generation synergy with two examples from HotPotQA and StrategyQA, respectively. In the first iteration, as both questions need multi-hop reasoning, the retriever fails to retrieve all supporting knowledge using only the questions. Despite being affected by distractive retrieved knowledge (*the capacity of a different arena* in the example from HotPotQA) and showing imperfect parametric knowledge (the generated statement that *Raclette is unlikely to be found in Paris* in the example from StrategyQA) in the first iteration, the LLM generates phrases that help retrieve relevant knowledge in the second iteration, and successfully corrects its outputs.

## 6 Conclusion

We demonstrate the effectiveness of ITER-RETGEN in answering questions with complex information needs. Despite simple, ITER-RETGEN outperforms retrieval-augmented methods that have a more complex workflow, which we believe could serve as a strong baseline for future research on retrieval-augmented generation. We also show that generation-augmented retrieval adaptation can further improve the performance of ITER-RETGEN while also reducing overheads.

## Limitations

In this work, we propose to enhance retrieval-augmented large language models with ITER-RETGEN which synergizes retrieval and generation in an iterative manner, and demonstrates strong performance compared to more structured prompting techniques such as Self-Ask. However, it's worth noting that our experiments utilized a fixed black-box large language model, which may not have been equally optimized for various forms of prompting. It would be intriguing to investigate the potential of prompting-specific (gradient-based) optimization in pushing the limits further. This could involve enabling a large language model to leverage parametric and non-parametric knowledge more flexibly and effectively. By exploring this avenue, we may uncover new insights and advancements in the field. Furthermore, our experiments did not cover long-form generation which would probably benefit from more fine-grained retrieval than ITER-RETGEN does in this work. We acknowledge that this area warrants further exploration, and we leave it for future work.

## Acknowledgements

Zhihong Shao and Minlie Huang were supported by the National Science Foundation for Distinguished Young Scholars (with No. 62125604) and the NSFC projects (Key project with No. 61936010). They were also supported by the Guoqiang Institute of Tsinghua University, with Grant No. 2020GQG0005.

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

## A Experiments Using Llama-2

To demonstrate the effectiveness of ITER-RETGEN on open-source models, we replaced the generation model `text-davinci-003` in Table 2 with Llama-2 models (Touvron et al., 2023), and re-ran the evaluation. As shown in Table 8, ITER-RETGEN consistently outperforms all baselines significantly.

## B Few-Shot Prompts

In this section, we present all few-shot prompts used in our experiments. We replace retrieved paragraphs with the placeholder {Knowledge} for brevity. CoT prompting shares the same in-context demonstrations with ITER-RETGEN, except that it is not augmented with retrieval.

### B.1 HotPotQA

Prompts for Direct Prompting, ReAct, Self-Ask, and ITER-RETGEN are presented in Table 9, Table 10, Table 11, and Table 12, respectively.

### B.2 2WikiMultiHopQA

Prompts for Direct Prompting, ReAct, Self-Ask, and ITER-RETGEN are presented in Table 13, Table 14, Table 15, and Table 16, respectively.

### B.3 MuSiQue

Prompts for Direct Prompting, ReAct, Self-Ask, and ITER-RETGEN are presented in Table 17, Table 18, Table 19, and Table 20, respectively.

### B.4 Bamboogle

Prompts for Direct Prompting, ReAct, Self-Ask, and ITER-RETGEN are presented in Table 21, Table 22, Table 23, and Table 24, respectively.

### B.5 Feverous

Prompts for Direct Prompting, ReAct, Self-Ask, and ITER-RETGEN are presented in Table 25, Table 26, Table 27, and Table 28, respectively.

### B.6 StrategyQA

Prompts for Direct Prompting, ReAct, Self-Ask, and ITER-RETGEN are presented in Table 29, Table 30, Table 31, and Table 32, respectively.

| Model | Llama-2-13B | | | Llama-2-70B | | |
|---|---|---|---|---|---|---|
| Dataset | HotPotQA | 2WikiMultiHopQA | StrategyQA | HotPotQA | 2WikiMultiHopQA | StrategyQA |
| Without Retrieval | | | | | | |
| Direct | 36.4 | 31.6 | 60.5 | 47.2 | 39.0 | 72.7 |
| CoT | 43.0 | 33.2 | 63.7 | 55.2 | 46.0 | 72.7 |
| With Retrieval | | | | | | |
| Direct | 51.8 | 38.6 | 63.3 | 58.6 | 45.1 | 73.3 |
| ReAct | 36.0 | 27.5 | 61.5 | 42.6 | 36.8 | 69.5 |
| Self-Ask | 45.8 | 38.5 | 63.3 | 58.4 | 53.2 | 71.7 |
| ITER-RETGEN 1 | 53.8 | 44.6 | 62.8 | 64.4 | 55.1 | 74.8 |
| ITER-RETGEN 2 | **57.8** | **48.0** | **67.2** | **67.8** | **57.9** | **76.6** |

Table 8: Experiments using the open-source Llama-2 models. We used $\text{Acc}^\dagger$ as the evaluation metric, i.e., to evaluate the accuracy of model outputs with `text-davinci-003`.

---

{Knowledge}
Question: What is the name of this American musician, singer, actor, comedian, and songwriter, who worked with Modern Records and born in December 5, 1932?
The answer is Little Richard

{Knowledge}
Question: Between Chinua Achebe and Rachel Carson, who had more diverse jobs?
The answer is Chinua Achebe

{Knowledge}
Question: Remember Me Ballin' is a CD single by Indo G that features an American rapper born in what year?
The answer is 1979

---

Table 9: 3-Shot Demonstrations for Direct Prompting on HotPotQA.

Given the following question, answer it by providing follow up questions and intermediate answers. For each follow up question, you are given a context which is the top returned Wikipedia snippets for the question. If no follow up questions are necessary, answer the question directly.
#
Question: What is the name of this American musician, singer, actor, comedian, and songwriter, who worked with Modern Records and born in December 5, 1932?
Are follow up questions needed here: Yes.
Follow up: Who worked with Modern Records?
{Knowledge}
Intermediate answer: Artists worked with Modern Records include Etta James, Little Richard, Joe Houston, Ike and Tina Turner and John Lee Hooker.
Follow up: Is Etta James an American musician, singer, actor, comedian, and songwriter, and was born in December 5, 1932?
{Knowledge}
Intermediate answer: Etta James was born in January 25, 1938, not December 5, 1932, so the answer is no.
Follow up: Is Little Richard an American musician, singer, actor, comedian, and songwriter, and was born in December 5, 1932?
{Knowledge}
Intermediate answer: Yes, Little Richard, born in December 5, 1932, is an American musician, singer, actor, comedian and songwriter.
So the final answer is: Little Richard
#
Question: Between Chinua Achebe and Rachel Carson, who had more diverse jobs?
Are follow up questions needed here: Yes.
Follow up: What jobs did Chinua Achebe have?
{Knowledge}
Intermediate answer: Chinua Achebe was a Nigerian (1) novelist, (2) poet, (3) professor, and (4) critic, so Chinua Achebe had 4 jobs.
Follow up: What jobs did Rachel Carson have?
{Knowledge}
Intermediate answer: Rachel Carson was an American (1) marine biologist, (2) author, and (3) conservationist, so Rachel Carson had 3 jobs.
Follow up: Did Chinua Achebe have more jobs than Rachel Carson?
{Knowledge}
Intermediate answer: Chinua Achebe had 4 jobs, while Rachel Carson had 3 jobs. 4 is greater than 3, so yes, Chinua Achebe had more jobs.
So the final answer is: Chinua Achebe
#
Question: Remember Me Ballin' is a CD single by Indo G that features an American rapper born in what year?
Are follow up questions needed here: Yes.
Follow up: Which American rapper is featured by Remember Me Ballin', a CD single by Indo G?
{Knowledge}
Intermediate answer: Gangsta Boo
Follow up: In which year was Gangsta Boo born?
{Knowledge}
Intermediate answer: Gangsta Boo was born in August 7, 1979, so the answer is 1979.
So the final answer is: 1979

Table 10: 3-Shot Demonstrations for ReAct on HotPotQA.

Given the following question, answer it by providing follow up questions and intermediate answers. For each follow up question, you are given a context which is the top returned Wikipedia snippets for the question. If no follow up questions are necessary, answer the question directly.
#
{Knowledge}
Question: What is the name of this American musician, singer, actor, comedian, and songwriter, who worked with Modern Records and born in December 5, 1932?
Are follow up questions needed here: Yes.
Follow up: Who worked with Modern Records?
Intermediate answer: Artists worked with Modern Records include Etta James, Little Richard, Joe Houston, Ike and Tina Turner and John Lee Hooker.
Follow up: Is Etta James an American musician, singer, actor, comedian, and songwriter, and was born in December 5, 1932?
Intermediate answer: Etta James was born in January 25, 1938, not December 5, 1932, so the answer is no.
Follow up: Is Little Richard an American musician, singer, actor, comedian, and songwriter, and was born in December 5, 1932?
Intermediate answer: Yes, Little Richard, born in December 5, 1932, is an American musician, singer, actor, comedian and songwriter.
So the final answer is: Little Richard
#
{Knowledge}
Question: Between Chinua Achebe and Rachel Carson, who had more diverse jobs?
Are follow up questions needed here: Yes.
Follow up: What jobs did Chinua Achebe have?
Intermediate answer: Chinua Achebe was a Nigerian (1) novelist, (2) poet, (3) professor, and (4) critic, so Chinua Achebe had 4 jobs.
Follow up: What jobs did Rachel Carson have?
Intermediate answer: Rachel Carson was an American (1) marine biologist, (2) author, and (3) conservationist, so Rachel Carson had 3 jobs.
Follow up: Did Chinua Achebe have more jobs than Rachel Carson?
Intermediate answer: Chinua Achebe had 4 jobs, while Rachel Carson had 3 jobs. 4 is greater than 3, so yes, Chinua Achebe had more jobs.
So the final answer is: Chinua Achebe
#
{Knowledge}
Question: Remember Me Ballin' is a CD single by Indo G that features an American rapper born in what year?
Are follow up questions needed here: Yes.
Follow up: Which American rapper is featured by Remember Me Ballin', a CD single by Indo G?
Intermediate answer: Gangsta Boo
Follow up: In which year was Gangsta Boo born?
Intermediate answer: Gangsta Boo was born in August 7, 1979, so the answer is 1979.
So the final answer is: 1979

Table 11: 3-Shot Demonstrations for Self-Ask on HotPotQA.

{Knowledge}
Question: What is the name of this American musician, singer, actor, comedian, and songwriter, who worked with Modern Records and born in December 5, 1932?
Let's think step by step.
Artists who worked with Modern Records include Etta James, Joe Houston, Little Richard, Ike and Tina Turner and John Lee Hooker in the 1950s and 1960s. Of these Little Richard, born in December 5, 1932, was an American musician, singer, actor, comedian, and songwriter.
So the answer is Little Richard

{Knowledge}
Question: Between Chinua Achebe and Rachel Carson, who had more diverse jobs?
Let's think step by step.
Chinua Achebe was a Nigerian novelist, poet, professor, and critic. Rachel Carson was an American marine biologist, author, and conservationist. So Chinua Achebe had 4 jobs, while Rachel Carson had 3 jobs. Chinua Achebe had more diverse jobs than Rachel Carson.
So the answer is Chinua Achebe

{Knowledge}
Question: Remember Me Ballin' is a CD single by Indo G that features an American rapper born in what year?
Let's think step by step.
Remember Me Ballin' is the CD single by Indo G featuring Gangsta Boo. Gangsta Boo is Lola Mitchell's stage name, who was born in August 7, 1979, and is an American rapper.
So the answer is 1979

Table 12: 3-Shot Demonstrations for ITER-RETGEN on HotPotQA.

{Knowledge}
Question: Which film came out first, Blind Shaft or The Mask Of Fu Manchu?
The answer is The Mask Of Fu Manchu

{Knowledge}
Question: When did John V, Prince Of Anhalt-Zerbst's father die?
The answer is 12 June 1516

{Knowledge}
Question: Which film has the director who was born later, El Extrano Viaje or Love In Pawn?
The answer is El Extrano Viaje

Table 13: 3-Shot Demonstrations for Direct Prompting on 2WikiMultiHopQA.

Given the following question, answer it by providing follow up questions and intermediate answers. For each follow up question, you are given a context which is the top returned Wikipedia snippets for the question. If no follow up questions are necessary, answer the question directly.
#
Question: Which film came out first, Blind Shaft or The Mask Of Fu Manchu?
Are follow up questions needed here: Yes.
Follow up: When did Blind Shaft come out?
{Knowledge}
Intermediate answer: Blind Shaft came out in 2003.
Follow up: When did The Mask Of Fu Manchu come out?
{Knowledge}
Intermediate answer: The Mask Of Fu Manchu came out in 1932.
So the final answer is: The Mask Of Fu Manchu
#
Question: When did John V, Prince Of Anhalt-Zerbst's father die?
Are follow up questions needed here: Yes.
Follow up: Who is the father of John V, Prince Of Anhalt-Zerbst?
{Knowledge}
Intermediate answer: The father of John V, Prince Of Anhalt-Zerbst is Ernest I, Prince of Anhalt-Dessau.
Follow up: When did Ernest I, Prince of Anhalt-Dessau die?
{Knowledge}
Intermediate answer: Ernest I, Prince of Anhalt-Dessau died on 12 June 1516.
So the final answer is: 12 June 1516
#
Question: Which film has the director who was born later, El Extrano Viaje or Love In Pawn?
Are follow up questions needed here: Yes.
Follow up: Who is the director of El Extrano Viaje?
{Knowledge}
Intermediate answer: The director of El Extrano Viaje is Fernando Fernan Gomez.
Follow up: Who is the director of Love in Pawn?
{Knowledge}
Intermediate answer: The director of Love in Pawn is Charles Saunders.
Follow up: When was Fernando Fernan Gomez born?
{Knowledge}
Intermediate answer: Fernando Fernan Gomez was born on 28 August 1921.
Follow up: When was Charles Saunders (director) born?
{Knowledge}
Intermediate answer: Charles Saunders was born on 8 April 1904.
So the final answer is: El Extrano Viaje

Table 14: 3-Shot Demonstrations for ReAct on 2WikiMultiHopQA.

Given the following question, answer it by providing follow up questions and intermediate answers. For each follow up question, you are given a context which is the top returned Wikipedia snippets for the question. If no follow up questions are necessary, answer the question directly.
#
{Knowledge}
Question: Which film came out first, Blind Shaft or The Mask Of Fu Manchu?
Are follow up questions needed here: Yes.
Follow up: When did Blind Shaft come out?
Intermediate answer: Blind Shaft came out in 2003.
Follow up: When did The Mask Of Fu Manchu come out?
Intermediate answer: The Mask Of Fu Manchu came out in 1932.
So the final answer is: The Mask Of Fu Manchu
#
{Knowledge}
Question: When did John V, Prince Of Anhalt-Zerbst's father die?
Are follow up questions needed here: Yes.
Follow up: Who is the father of John V, Prince Of Anhalt-Zerbst?
Intermediate answer: The father of John V, Prince Of Anhalt-Zerbst is Ernest I, Prince of Anhalt-Dessau.
Follow up: When did Ernest I, Prince of Anhalt-Dessau die?
Intermediate answer: Ernest I, Prince of Anhalt-Dessau died on 12 June 1516.
So the final answer is: 12 June 1516
#
{Knowledge}
Question: Which film has the director who was born later, El Extrano Viaje or Love In Pawn?
Are follow up questions needed here: Yes.
Follow up: Who is the director of El Extrano Viaje?
Intermediate answer: The director of El Extrano Viaje is Fernando Fernan Gomez.
Follow up: Who is the director of Love in Pawn?
Intermediate answer: The director of Love in Pawn is Charles Saunders.
Follow up: When was Fernando Fernan Gomez born?
Intermediate answer: Fernando Fernan Gomez was born on 28 August 1921.
Follow up: When was Charles Saunders (director) born?
Intermediate answer: Charles Saunders was born on 8 April 1904.
So the final answer is: El Extrano Viaje

Table 15: 3-Shot Demonstrations for Self-Ask on 2WikiMultiHopQA.

{Knowledge}
Question: Which film came out first, Blind Shaft or The Mask Of Fu Manchu?
Let's think step by step.
Blind Shaft is a 2003 film, while The Mask Of Fu Manchu opened in New York on December 2, 1932. 2003 comes after 1932. Therefore, The Mask Of Fu Manchu came out earlier than Blind Shaft.
So the answer is The Mask Of Fu Manchu

{Knowledge}
Question: When did John V, Prince Of Anhalt-Zerbst's father die?
Let's think step by step.
John was the second son of Ernest I, Prince of Anhalt-Dessau. Ernest I, Prince of Anhalt-Dessau died on 12 June 1516.
So the answer is 12 June 1516

{Knowledge}
Question: Which film has the director who was born later, El Extrano Viaje or Love In Pawn?
Let's think step by step.
The director of El Extrano Viaje is Fernando Fernan Gomez, who was born on 28 August 1921. The director of Love In Pawn is Charles Saunders, who was born on 8 April 1904. 28 August 1921 comes after 8 April 1904. Therefore, Fernando Fernan Gomez was born later than Charles Saunders.
So the answer is El Extrano Viaje

Table 16: 3-Shot Demonstrations for ITER-RETGEN on 2WikiMultiHopQA.

{Knowledge}
Question: In which year did the publisher of In Cold Blood form?
The answer is 2001

{Knowledge}
Question: Who was in charge of the city where The Killing of a Sacred Deer was filmed?
The answer is John Cranley

{Knowledge}
Question: Where on the Avalon Peninsula is the city that Signal Hill overlooks?
The answer is eastern tip

Table 17: 3-Shot Demonstrations for Direct Prompting on MuSiQue.

Given the following question, answer it by providing follow up questions and intermediate answers. For each follow up question, you are given a context which is the top returned Wikipedia snippets for the question. If no follow up questions are necessary, answer the question directly.
#
Question: In which year did the publisher of In Cold Blood form?
Are follow up questions needed here: Yes.
Follow up: What business published In Cold Blood?
{Knowledge}
Intermediate answer: In Cold Blood was published in book form by Random House.
Follow up: Which year witnessed the formation of Random House?
{Knowledge}
Intermediate answer: Random House was form in 2001.
So the final answer is: 2001
#
Question: Who was in charge of the city where The Killing of a Sacred Deer was filmed?
Are follow up questions needed here: Yes.
Follow up: In which city was The Killing of a Sacred Deer filmed
{Knowledge}
Intermediate answer: The Killing of a Sacred Deer was filmed in Cincinnati.
Follow up: Who was in charge of Cincinnati?
{Knowledge}
Intermediate answer: The present Mayor of Cincinnati is John Cranley, so John Cranley is in charge.
So the final answer is: John Cranley
#
Question: Where on the Avalon Peninsula is the city that Signal Hill overlooks?
Are follow up questions needed here: Yes.
Follow up: What city does Signal Hill overlook?
{Knowledge}
Intermediate answer: Signal Hill is a hill which overlooks the city of St. John's.
Follow up: Where on the Avalon Peninsula is St. John's located?
{Knowledge}
Intermediate answer: St. John's is located on the eastern tip of the Avalon Peninsula.
So the final answer is: eastern tip

Table 18: 3-Shot Demonstrations for ReAct on MuSiQue.

Given the following question, answer it by providing follow up questions and intermediate answers. For each follow up question, you are given a context which is the top returned Wikipedia snippets for the question. If no follow up questions are necessary, answer the question directly.
#
{Knowledge}
Question: In which year did the publisher of In Cold Blood form?
Are follow up questions needed here: Yes.
Follow up: What business published In Cold Blood?
Intermediate answer: In Cold Blood was published in book form by Random House.
Follow up: Which year witnessed the formation of Random House?
Intermediate answer: Random House was form in 2001.
So the final answer is: 2001
#
{Knowledge}
Question: Who was in charge of the city where The Killing of a Sacred Deer was filmed?
Are follow up questions needed here: Yes.
Follow up: In which city was The Killing of a Sacred Deer filmed
Intermediate answer: The Killing of a Sacred Deer was filmed in Cincinnati.
Follow up: Who was in charge of Cincinnati?
Intermediate answer: The present Mayor of Cincinnati is John Cranley, so John Cranley is in charge.
So the final answer is: John Cranley
#
{Knowledge}
Question: Where on the Avalon Peninsula is the city that Signal Hill overlooks?
Are follow up questions needed here: Yes.
Follow up: What city does Signal Hill overlook?
Intermediate answer: Signal Hill is a hill which overlooks the city of St. John's.
Follow up: Where on the Avalon Peninsula is St. John's located?
Intermediate answer: St. John's is located on the eastern tip of the Avalon Peninsula.
So the final answer is: eastern tip

Table 19: 3-Shot Demonstrations for Self-Ask on MuSiQue.

{Knowledge}
Question: In which year did the publisher of In Cold Blood form?
Let's think step by step.
In Cold Blood was first published in book form by Random House. Random House was form in 2001.
So the answer is 2001

{Knowledge}
Question: Who was in charge of the city where The Killing of a Sacred Deer was filmed?
Let's think step by step.
The Killing of a Sacred Deer was filmed in Cincinnati. The present Mayor of Cincinnati is John Cranley. Therefore, John Cranley is in charge of the city.
So the answer is John Cranley

{Knowledge}
Question: Where on the Avalon Peninsula is the city that Signal Hill overlooks?
Let's think step by step.
Signal Hill is a hill which overlooks the city of St. John's. St. John's is located on the eastern tip of the Avalon Peninsula.
So the answer is eastern tip

Table 20: 3-Shot Demonstrations for ITER-RETGEN on MuSiQue.

{Knowledge}
Question: When did the first prime minister of the Russian Empire come into office?
The answer is 1905-11-06 00:00:00

{Knowledge}
Question: The most populous city in Punjab is how large (area wise)?
The answer is 310 square kilometers

{Knowledge}
Question: What is the capital of the country where yoga originated?
The answer is New Delhi

Table 21: 3-Shot Demonstrations for Direct Prompting on Bamboogle.

Given the following question, answer it by providing follow up questions and intermediate answers. For each follow up question, you are given a context which is the top returned Wikipedia snippets for the question. If no follow up questions are necessary, answer the question directly.
#
Question: When did the first prime minister of the Russian Empire come into office?
Are follow up questions needed here: Yes.
Follow up: Who is the first prime minister of the Russian Empire?
{Knowledge}
Intermediate answer: Sergei Witte
Follow up: When did Sergei Witte come into office?
{Knowledge}
Intermediate answer: Sergei Witte was appointed on 6 November 1905.
So the final answer is: 1905-11-06 00:00:00
#
Question: The most populous city in Punjab is how large (area wise)?
Are follow up questions needed here: Yes.
Follow up: What is the most populous city in Punjab?
{Knowledge}
Intermediate answer: Ludhiana is the most populous and largest city in Punjab.
Follow up: How large is Ludhiana, the most populous city in Punjab?
{Knowledge}
Intermediate answer: The area of Ludhiana is over 310 km2.
So the final answer is: 310 square kilometers
#
Question: What is the capital of the country where yoga originated?
Are follow up questions needed here: Yes.
Follow up: Which country was yoga originated?
{Knowledge}
Intermediate answer: There is no consensus on yoga's origin. Suggested origins include India.
Follow up: What is the capital of India?
{Knowledge}
Intermediate answer: The current capital of India is New Delhi.
So the final answer is: New Delhi

Table 22: 3-Shot Demonstrations for ReAct on Bamboogle.

Given the following question, answer it by providing follow up questions and intermediate answers. For each follow up question, you are given a context which is the top returned Wikipedia snippets for the question. If no follow up questions are necessary, answer the question directly.
#
{Knowledge}
Question: When did the first prime minister of the Russian Empire come into office?
Are follow up questions needed here: Yes.
Follow up: Who is the first prime minister of the Russian Empire?
Intermediate answer: Sergei Witte
Follow up: When did Sergei Witte come into office?
Intermediate answer: Sergei Witte was appointed on 6 November 1905.
So the final answer is: 1905-11-06 00:00:00
#
{Knowledge}
Question: The most populous city in Punjab is how large (area wise)?
Are follow up questions needed here: Yes.
Follow up: What is the most populous city in Punjab?
Intermediate answer: Ludhiana is the most populous and largest city in Punjab.
Follow up: How large is Ludhiana, the most populous city in Punjab?
Intermediate answer: The area of Ludhiana is over 310 km2.
So the final answer is: 310 square kilometers
#
{Knowledge}
Question: What is the capital of the country where yoga originated?
Are follow up questions needed here: Yes.
Follow up: Which country was yoga originated?
Intermediate answer: There is no consensus on yoga's origin. Suggested origins include India.
Follow up: What is the capital of India?
Intermediate answer: The current capital of India is New Delhi.
So the final answer is: New Delhi

Table 23: 3-Shot Demonstrations for Self-Ask on Bamboogle.

{Knowledge}
Question: When did the first prime minister of the Russian Empire come into office?
Let's think step by step.
The first prime minister of the Russian Empire was Count Sergei Witte. Sergei Witte was appointed on 6 November 1905.
So the answer is 1905-11-06 00:00:00

{Knowledge}
Question: The most populous city in Punjab is how large (area wise)?
Let's think step by step.
Ludhiana is the most populous and the largest city in the Indian state of Punjab. The city has an area of over 310 km2.
So the answer is 310 square kilometers

{Knowledge}
Question: What is the capital of the country where yoga originated?
Let's think step by step.
Suggested origins include pre-Vedic Eastern states of India. The current capital of India is New Delhi.
So the answer is New Delhi

Table 24: 3-Shot Demonstrations for ITER-RETGEN on Bamboogle.

{Knowledge}
Question: Is it true that Belgrade Race is an annual men's footrace of around 6 kilometres (5834 metres) that is held in Belgrade, Serbia through history, past winners includes Brahim Lahlafi (1st edition), Philip Mosima (3rd) and Josphat Menjo (6th)?
The answer is Yes

{Knowledge}
Question: Is it true that Based on the same platform as the Chevrolet Sail, the Baojun 310 was launched on 2017 Beijing Auto Show where the price ranges from 36.800 yuan to 60.800 yuan?
The answer is No

{Knowledge}
Question: Is it true that Florida International University pedestrian bridge collapse was funded with a $19.4 million Transportation Investment Generating Economic Recovery grant from the United States Department of Transportation in 2013, along with state agencies and the bridge cost $14.2 million to construct?
The answer is No

Table 25: 3-Shot Demonstrations for Direct Prompting on Feverous.

Given the following question, answer it by providing follow up questions and intermediate answers. For each follow up question, you are given a context which is the top returned Wikipedia snippets for the question. If no follow up questions are necessary, answer the question directly. The final answer should always be either Yes or No, and NOTHING ELSE.
#
Question: Is it true that Belgrade Race is an annual men's footrace of around 6 kilometres (5834 metres) that is held in Belgrade, Serbia through history, past winners includes Brahim Lahlafi (1st edition), Philip Mosima (3rd) and Josphat Menjo (6th)?
Are follow up questions needed here: Yes.
Follow up: What is the Belgrade Race?
{Knowledge}
Intermediate answer: The Belgrade Race Through History is an annual men's footrace of around 6 kilometres (5834 metres) that is held in Belgrade, Serbia.
Follow up: Has Brahim Lahlafi won Belgrade Race?
{Knowledge}
Intermediate answer: Yes, Brahim Lahlafi was the winner in 1996.
Follow up: Has Philip Mosima won Belgrade Race?
{Knowledge}
Intermediate answer: Yes, Philip Mosima beat Marathon world record and won in 1998
Follow up: Has Josphat Menjo won Belgrade Race?
{Knowledge}
Intermediate answer: Yes, Josphat Menjo broke the meet record and won the competition.
So the final answer is: Yes
#
Question: Is it true that Based on the same platform as the Chevrolet Sail, the Baojun 310 was launched on 2017 Beijing Auto Show where the price ranges from 36.800 yuan to 60.800 yuan?
Are follow up questions needed here: Yes.
Follow up: When and where was the Baojun 310 launched?
{Knowledge}
Intermediate answer: The Baojun 310 was launched on 2016 Beijing Auto Show, not 2017 Beijing Auto Show.
So the final answer is: No
#
Question: Is it true that Florida International University pedestrian bridge collapse was funded with a $19.4 million Transportation Investment Generating Economic Recovery grant from the United States Department of Transportation in 2013, along with state agencies and the bridge cost $14.2 million to construct?
Are follow up questions needed here: Yes.
Follow up: How was Florida International University pedestrian bridge collapse funded?
{Knowledge}
Intermediate answer: Florida International University pedestrian bridge was a $14.2 million project funded with a $19.4 million Transportation Investment Generating Economic Recovery (TIGER) grant from the United States Department of Transportation in 2013, along with state agencies, which is consistent with facts in the question.
Follow up: How much did it cost to construct Florida International University pedestrian bridge?
{Knowledge}
Intermediate answer: The bridge cost $9 million to construct, not $14.2 million.
So the final answer is: No

Table 26: 3-Shot Demonstrations for ReAct on Feverous.

Given the following question, answer it by providing follow up questions and intermediate answers. For each follow up question, you are given a context which is the top returned Wikipedia snippets for the question. If no follow up questions are necessary, answer the question directly. The final answer should always be either Yes or No, and NOTHING ELSE.
#
{Knowledge}
Question: Is it true that Belgrade Race is an annual men's footrace of around 6 kilometres (5834 metres) that is held in Belgrade, Serbia through history, past winners includes Brahim Lahlafi (1st edition), Philip Mosima (3rd) and Josphat Menjo (6th)?
Are follow up questions needed here: Yes.
Follow up: What is the Belgrade Race?
Intermediate answer: The Belgrade Race Through History is an annual men's footrace of around 6 kilometres (5834 metres) that is held in Belgrade, Serbia.
Follow up: Has Brahim Lahlafi won Belgrade Race?
Intermediate answer: Yes, Brahim Lahlafi was the winner in 1996.
Follow up: Has Philip Mosima won Belgrade Race?
Intermediate answer: Yes, Philip Mosima beat Marathon world record and won in 1998
Follow up: Has Josphat Menjo won Belgrade Race?
Intermediate answer: Yes, Josphat Menjo broke the meet record and won the competition.
So the final answer is: Yes
#
{Knowledge}
Question: Is it true that Based on the same platform as the Chevrolet Sail, the Baojun 310 was launched on 2017 Beijing Auto Show where the price ranges from 36.800 yuan to 60.800 yuan?
Are follow up questions needed here: Yes.
Follow up: When and where was the Baojun 310 launched?
Intermediate answer: The Baojun 310 was launched on 2016 Beijing Auto Show, not 2017 Beijing Auto Show.
So the final answer is: No
#
{Knowledge}
Question: Is it true that Florida International University pedestrian bridge collapse was funded with a $19.4 million Transportation Investment Generating Economic Recovery grant from the United States Department of Transportation in 2013, along with state agencies and the bridge cost $14.2 million to construct?
Are follow up questions needed here: Yes.
Follow up: How was Florida International University pedestrian bridge collapse funded?
Intermediate answer: Florida International University pedestrian bridge was a $14.2 million project funded with a $19.4 million Transportation Investment Generating Economic Recovery (TIGER) grant from the United States Department of Transportation in 2013, along with state agencies, which is consistent with facts in the question.
Follow up: How much did it cost to construct Florida International University pedestrian bridge?
Intermediate answer: The bridge cost $9 million to construct, not $14.2 million.
So the final answer is: No

Table 27: 3-Shot Demonstrations for Self-Ask on Feverous.

You are required to verify facts in the following questions. The final answer to a question should always be either Yes or No, and NOTHING ELSE.

{Knowledge}
Question: Is it true that Belgrade Race is an annual men's footrace of around 6 kilometres (5834 metres) that is held in Belgrade, Serbia through history, past winners includes Brahim Lahlafi (1st edition), Philip Mosima (3rd) and Josphat Menjo (6th)?
Let's think step by step.
I need to verify facts in the question. The Belgrade Race Through History is an annual men's footrace of around 6 kilometres (5834 metres) that is held in Belgrade, Serbia. In 1996 Brahim Lahlafi was the winner of the competition. Philip Mosima won the competition in 1998, and beat Marathon world record holder Paul Tergat. Josphat Menjo also won the competition and broke the meet record. Therefore, past winners include Brahim Lahlafi, Philip Mosima and Josphat Menjo. All facts are verified.
So the answer is Yes

{Knowledge}
Question: Is it true that Based on the same platform as the Chevrolet Sail, the Baojun 310 was launched on 2017 Beijing Auto Show where the price ranges from 36.800 yuan to 60.800 yuan?
Let's think step by step.
I need to verify facts in the question. The Baojun 310 was indeed based on the same platform as the Chevrolet Sail. The Baojun 310 was launched on 2016 Beijing Auto Show, not 2017 Beijing Auto Show.
So the answer is No

{Knowledge}
Question: Is it true that Florida International University pedestrian bridge collapse was funded with a $19.4 million Transportation Investment Generating Economic Recovery grant from the United States Department of Transportation in 2013, along with state agencies and the bridge cost $14.2 million to construct?
Let's think step by step.
I need to verify facts in the question. Florida International University pedestrian bridge was a $14.2 million project funded with a $19.4 million Transportation Investment Generating Economic Recovery (TIGER) grant from the United States Department of Transportation in 2013, along with state agencies. The Bridge cost $8 million to construct, not $14.2 million.
So the answer is No

Table 28: 3-Shot Demonstrations for ITER-RETGEN on Feverous.

{Knowledge}
Question: Is it common to see frost during some college commencements?
The answer is Yes

{Knowledge}
Question: Could a llama birth twice during War in Vietnam (1945-46)?
The answer is No

{Knowledge}
Question: Would Richard Dawkins hypothetically refuse an offering of the Last rites?
The answer is Yes

Table 29: 3-Shot Demonstrations for Direct Prompting on StrategyQA.

Given the following question, answer it by providing follow up questions and intermediate answers. For each follow up question, you are given a context which is the top returned Wikipedia snippets for the question. If no follow up questions are necessary, answer the question directly. The final answer should always be either Yes or No, and NOTHING ELSE.
#
Question: Is it common to see frost during some college commencements?
Are follow up questions needed here: Yes.
Follow up: What seasons can you expect to see frost?
{Knowledge}
Intermediate answer: Frost is common during the winter.
Follow up: When is college commencement?
{Knowledge}
Intermediate answer: College commencement ceremonies often happen during the months of December, May, June.
Follow up: Do any of the months December, May, June occur during the Winter?
{Knowledge}
Intermediate answer: December is in the winter.
So the final answer is: Yes
#
Question: Could a llama birth twice during War in Vietnam (1945-46)?
Are follow up questions needed here: Yes.
Follow up: How long did the vietnam war (1945-1946) last?
{Knowledge}
Intermediate answer: The War in Vietnam (1945-46) lasted around 6 months.
Follow up: How long is the llama gestation period?
{Knowledge}
Intermediate answer: The gestation period for a llama is 11.5 months.
Follow up: What is 2 times 11.5?
{Knowledge}
Intermediate answer: 23, which is longer than 6.
So the final answer is: No
#
Question: Would Richard Dawkins hypothetically refuse an offering of the Last rites?
Are follow up questions needed here: Yes.
Follow up: What are the last Rites?
{Knowledge}
Intermediate answer: The Last rites, in Catholicism, are the last prayers and ministrations given to an individual of the faith, when possible, shortly before death.
Follow up: What are Richard Dawkins religious beliefs?
{Knowledge}
Intermediate answer: Richard Dawkins is known as an outspoken atheist, well known for his criticism of creationism and intelligent design.
Follow up: Would an atheist participate in Catholics prayers?
{Knowledge}
Intermediate answer: It is unlikely that an atheist would participate in Catholics prayers.
So the final answer is: Yes

Table 30: 3-Shot Demonstrations for ReAct on StrategyQA.

Given the following question, answer it by providing follow up questions and intermediate answers. For each follow up question, you are given a context which is the top returned Wikipedia snippets for the question. If no follow up questions are necessary, answer the question directly. The final answer should always be either Yes or No, and NOTHING ELSE.
#
{Knowledge}
Question: Is it common to see frost during some college commencements?
Are follow up questions needed here: Yes.
Follow up: What seasons can you expect to see frost?
Intermediate answer: Frost is common during the winter.
Follow up: When is college commencement?
Intermediate answer: College commencement ceremonies often happen during the months of December, May, June.
Follow up: Do any of the months December, May, June occur during the Winter?
Intermediate answer: December is in the winter.
So the final answer is: Yes
#
{Knowledge}
Question: Could a llama birth twice during War in Vietnam (1945-46)?
Are follow up questions needed here: Yes.
Follow up: How long did the vietnam war (1945-1946) last?
Intermediate answer: The War in Vietnam (1945-46) lasted around 6 months.
Follow up: How long is the llama gestation period?
Intermediate answer: The gestation period for a llama is 11.5 months.
Follow up: What is 2 times 11.5?
Intermediate answer: 23, which is longer than 6.
So the final answer is: No
#
{Knowledge}
Question: Would Richard Dawkins hypothetically refuse an offering of the Last rites?
Are follow up questions needed here: Yes.
Follow up: What are the last Rites?
Intermediate answer: The Last rites, in Catholicism, are the last prayers and ministrations given to an individual of the faith, when possible, shortly before death.
Follow up: What are Richard Dawkins religious beliefs?
Intermediate answer: Richard Dawkins is known as an outspoken atheist, well known for his criticism of creationism and intelligent design.
Follow up: Would an atheist participate in Catholics prayers?
Intermediate answer: It is unlikely that an atheist would participate in Catholics prayers.
So the final answer is: Yes

Table 31: 3-Shot Demonstrations for Self-Ask on StrategyQA.

You are required to answer the following questions. The final answer to a question should always be either Yes or No, and NOTHING ELSE.

{Knowledge}
Question: Is it common to see frost during some college commencements?
Let's think step by step.
College commencement ceremonies often happen during the months of December, May, and sometimes June. Frost isn't uncommon to see during the month of December, as it is the winter.
So the answer is Yes

{Knowledge}
Question: Could a llama birth twice during War in Vietnam (1945-46)?
Let's think step by step.
The War in Vietnam (1945-46) lasted around 6 months. The gestation period for a llama is 11 months. If a llama birth twice, the minimum time needed is 2 times 11 months, which is 22 months, longer than 6 months.
So the answer is No

{Knowledge}
Question: Would Richard Dawkins hypothetically refuse an offering of the Last rites?
Let's think step by step.
Richard Dawkins is known as an outspoken atheist, well known for his criticism of creationism and intelligent design. The Last rites, in Catholicism, are the last prayers and ministrations given to an individual of the faith, when possible, shortly before death. It is unlikely that an atheist would participate in Catholics prayers.
So the answer is Yes

Table 32: 3-Shot Demonstrations for ITER-RETGEN on StrategyQA.