# OpenReview forum: "Enhancing Retrieval-Augmented Large Language Models with Iterative Retrieval-Generation Synergy"
_EMNLP/2023/Conference — EMNLP 2023 Findings_

### Official Review · Reviewer_ij2Q · 2023-07-30

**Soundness:** 4

**Excitement:**

3: Ambivalent: It has merits (e.g., it reports state-of-the-art results, the idea is nice), but there are key weaknesses (e.g., it describes incremental work), and it can significantly benefit from another round of revision. However, I won't object to accepting it if my co-reviewers champion it.

**Paper Topic And Main Contributions:**

This paper proposes an iterative retrieval-generation synergy framework for large language model and shows improved performance in question answering, fact verification and commonsense reasoning tasks.

**Reasons To Accept:**

- The present paper demonstrates strong motivation and brings up a crucial and intriguing question in the realm of LLM.
- With its well-crafted writing and descriptive figures, this paper effectively conveys its message.
- The paper offers a comprehensive set of experiments complemented by an extensive analysis that covers various aspects in detail.

**Reasons To Reject:**

- The concept presented in this paper is rather straightforward and seemingly trivial. The iterative retrieval approach, which uses the output from the previous round as input in the current retrieval process, is not a novel idea, and the paper does not highlight any unique aspects related to LLM. The study primarily applies this concept to retrieval-augmented LLM research with minor modifications, even lacking a heuristic solution to determine the number of iterative rounds.
- A notable omission in the paper is the absence of a comparison with the DSP framework, which would be the most appropriate baseline for this study.
- The experiments conducted solely utilize the text-davinci-003 model, with no accessible open-sourced models included. Moreover, the paper lacks any discussion concerning potential data leakage in the tested benchmarks when using text-davinci-003.

**Reproducibility:**

3: Could reproduce the results with some difficulty. The settings of parameters are underspecified or subjectively determined; the training/evaluation data are not widely available.

**Reviewer Confidence:**

4: Quite sure. I tried to check the important points carefully. It's unlikely, though conceivable, that I missed something that should affect my ratings.

---

> ### Author Rebuttal · Authors · 2023-08-29
>
> Thanks for your time, insightful comments, and questions. We provided our responses below.
>
> > The concept presented in this paper is rather straightforward and seemingly trivial. The iterative retrieval approach, which uses the output from the previous round as input in the current retrieval process, is not a novel idea, and the paper does not highlight any unique aspects related to LLM. The study primarily applies this concept to retrieval-augmented LLM research with minor modifications, even lacking a heuristic solution to determine the number of iterative rounds.
>
> We demonstrated in this paper that, Iter-RetGen, despite being simple, obtains strong results on various knowledge-intensive tasks, outperforming state-of-the-art methods including ReAct and Self-Ask while causing fewer overheads of both retrieval and generation. We also explored how we can further adapt the retriever with model generations, and achieved significantly better results.
>
> An important aspect related to LLMs is whether they can leverage both intrinsic and extrinsic knowledge flexibly and effectively. We showed that Iter-RetGen performs significantly better, regardless of whether in-context extrinsic knowledge mentions the answers or not.
>
> As for how to determine the number iterations, we showed in Table 2 that the second iteration usually gives the greatest performance boost, and already achieves significantly better or competitive results compared with baselines. Therefore, we recommend using 2 to 3 iterations. With the number of iterations increasing, performance improves more slowly, and overheads increase, too.
>
> > A notable omission in the paper is the absence of a comparison with the DSP framework, which would be the most appropriate baseline for this study.
>
> To compare with the method in the DSP paper, we used the [code](https://github.com/stanfordnlp/dspy/issues/85) shared by the author. We made the following adjustments for fair comparisons:
>
> * We changed DSP to use the same (1) model (i.e., text-davinci-003), (2) retrieval system (i.e., Contriever-MSMARCO on Wikipedia 2017 for HotPotQA), (3) number of paragraphs retrieved for each query (i.e., 5), (4) in-context QA pairs (for demonstration), and (5) the evaluation set (i.e., the first 500 questions from the dev set of HotPotQA) as in our paper.
> * The original DSP method produces an answer using sampling followed by majority voting. We changed DSP to use greedy decoding to produce the answer.
>
> Here are the experimental results on HotPotQA.
>
> | Method            | EM       | F1       | Acc      |
> | ----------------- | -------- | -------- | -------- |
> | Direct            | 31.6     | 44.7     | 53.3     |
> | ReAct             | 24.9     | 44.7     | 61.1     |
> | Self-Ask          | 36.8     | 55.2     | 64.8     |
> | DSP               | 43.8     | 55.0     | 60.8     |
> | Iter-RetGen (T=2) | **44.1** | **58.6** | **71.2** |
>
> Iter-RetGen outperforms DSP in terms of the three metrics.
>
> > The experiments conducted solely utilize the text-davinci-003 model, with no accessible open-sourced models included. Moreover, the paper lacks any discussion concerning potential data leakage in the tested benchmarks when using text-davinci-003.
>
> It would be interesting but challenging to measure data leakage without access to the training data. However, we believe Llama2 should be much less prone to data leakage than text-davinci-003, if such a concern exists.
>
> We experimented with the open-sourced Llama2-13B and Llama2-70B, without changing anything else. The accuracies of model predictions are presented as follows:
>
> | Model                  | Llama2-13B |                 |            | Llama2-70B |                 |            |
> | ---------------------- | ---------- | --------------- | ---------- | ---------- | --------------- | ---------- |
> | Dataset                | HotPotQA   | 2WikiMultiHopQA | StrategyQA | HotPotQA   | 2WikiMultiHopQA | StrategyQA |
> | Direct (w/o retrieval) | 36.4       | 31.6            | 60.5       | 47.2       | 39.0            | 72.7       |
> | CoT (w/o retrieval)    | 43.0       | 33.2            | 63.7       | 55.2       | 46.0            | 72.7       |
> |                        |            |                 |            |            |                 |            |
> | Direct                 | 51.8       | 38.6            | 63.3       | 58.6       | 45.1            | 73.3       |
> | ReAct                  | 36.0       | 27.5            | 61.5       | 42.6       | 36.8            | 69.5       |
> | Self-Ask               | 45.8       | 38.5            | 63.3       | 58.4       | 53.2            | 71.7       |
> |                        |            |                 |            |            |                 |            |
> | Iter-RetGen (T=1)      | 53.8       | 44.6            | 62.8       | 64.4       | 55.1            | 74.8       |
> | Iter-RetGen (T=2)      | **57.8**   | **48.0**        | **67.2**   | **67.8**   | **57.9**        | **76.6**   |
>
> As can be seen in the table, Iter-RetGen consistently outperforms all baselines significantly.

---

### Official Review · Reviewer_yJLF · 2023-08-04

**Soundness:** 3

**Excitement:**

3: Ambivalent: It has merits (e.g., it reports state-of-the-art results, the idea is nice), but there are key weaknesses (e.g., it describes incremental work), and it can significantly benefit from another round of revision. However, I won't object to accepting it if my co-reviewers champion it.

**Missing References:**

The following shows the similar process of distilling reranker to retrieval (Section 3.4). As such, the authors should cite this reference.

[1] Michael Glass, Gaetano Rossiello, Md Faisal Mahbub Chowdhury, Ankita Naik, Pengshan Cai, and Alfio Gliozzo. 2022. Re2G: Retrieve, Rerank, Generate. In Proceedings of the 2022 Conference of the North American Chapter of the Association for Computational Linguistics: Human Language Technologies, pages 2701–2715, Seattle, United States. Association for Computational Linguistics.

**Paper Topic And Main Contributions:**

The paper proposes ITER-RETGEN, which synergies retrieval and generation in an iterative manner. The process starts with a generation output, then concatenate the generated output with the retrieval result to generate next generated output. The process repeats until we reach a maximum number of iterations. The authors evaluate the proposed method on different tasks and datasets and show the effectiveness of ITER-RETGEN over Self-Ask and other baselines.

**Questions For The Authors:**

Q1. Can you elaborate how the previous methods "fail to process all the retrieved knowledge"? And how your method can achieve it?
Q2. Can you provide some insights on how the performance of ITER-GEn compared to Re2G? That is to say, we finetune the generation based on the retrieval outputs instead of using in-context learning?

Updated: I like to thank the authors for their feedbacks. However, the reply does not clearly address Q1, and since the novelty is not the strongest point of the paper, I like to keep my scores as it is.

**Reasons To Accept:**

1. The method is simple yet effective in a number of tasks and datasets.
2. The experimental design in 4.6.2 is interesting. It helps shed light of how the proposed method leverage parametric and non-parametric knowledge.
3. The authors propose a new metric based on LLM for evaluation.

**Reasons To Reject:**

1. Given that the method is simple and the idea of interleaving between retrieval and generation is not new, the novelty of this paper might be limited.
2. The work is motivated by the issues presented from line 063-070: (1) previous methods fail to process all the retrieved knowledge; (2) previous studies invoke retrieval multiple times. However, it is not clear to me how such issues affect the method design. For example, how ITER-GEN can "process all the retrieved knowledge"? In addition, theoretically, ITER-GEN also needs to invoke retrieval multiple times.
3. The improvement of retrieval is mostly due to the distillation process, not by the inclusion of y1. However, the method to distill the reranker to retrieval has been proposed in Re2G [1].

[1] Michael Glass, Gaetano Rossiello, Md Faisal Mahbub Chowdhury, Ankita Naik, Pengshan Cai, and Alfio Gliozzo. 2022. Re2G: Retrieve, Rerank, Generate. In Proceedings of the 2022 Conference of the North American Chapter of the Association for Computational Linguistics: Human Language Technologies, pages 2701–2715, Seattle, United States. Association for Computational Linguistics.

**Reproducibility:**

4: Could mostly reproduce the results, but there may be some variation because of sample variance or minor variations in their interpretation of the protocol or method.

**Reviewer Confidence:**

4: Quite sure. I tried to check the important points carefully. It's unlikely, though conceivable, that I missed something that should affect my ratings.

---

> ### Author Rebuttal · Authors · 2023-08-29
>
> Thanks for your time, insightful comments, and questions. We provided our responses below.
>
> > The work is motivated by the issues presented from line 063-070: (1) previous methods fail to process all the retrieved knowledge; (2) previous studies invoke retrieval multiple times. However, it is not clear to me how such issues affect the method design. For example, how ITER-GEN can "process all the retrieved knowledge"? In addition, theoretically, ITER-GEN also needs to invoke retrieval multiple times.
> >
> > Can you elaborate how the previous methods "fail to process all the retrieved knowledge"? And how your method can achieve it?
>
> The limitations we mentioned apply to the structured workflows adopted by previous work like ReAct and Self-Ask.
>
> * Limitation 1: `they fail to process all retrieved knowledge as a whole during the generation process`
>
>   * Methods like ReAct and Self-Ask interleave retrieval and generating intermediate reasoning steps, until the model generates the final answer. Each intermediate reasoning step is conditioned on knowledge retrieved before, with no awareness of knowledge retrieved afterwards.
>   * **How does Iter-RetGen alleviate this limitation?** In each iteration, Iter-RetGen retrieves knowledge based on (1) the question and (2) the previous model output which shows what may be needed to answer the question. Unlike ReAct and Self-Ask conditioning each reasoning step on partial knowledge, Iter-RetGen generates reasoning steps and the final answer based on all knowledge retrieved in an iteration. Notably, knowledge retrieved in an iteration is not complementary to that in the previous iteration, but tends to be more comprehensive and relevant. In our preliminary experiments, we have tried conditioning reasoning steps and the final answer on a merged version of knowledge from different iterations, but found no noticeable improvements.
>
> * Limitation 2: `they invoke retrieval multiple times, and may frequently change the prompts by updating newly retrieved knowledge, thus increasing the overheads of both retrieval and generation`
>
>   * To finalize a single answer, methods like ReAct and Self-Ask have to invoke multiple rounds of retrieval and generation to gather a comprehensive set of knowledge, with different rounds focusing on complementary aspects and updating the prompt accordingly.
>   * **How does Iter-RetGen alleviate this limitation?** Unlike ReAct and Self-Ask invoking multiple rounds of retrieval and generation to finalize a single answer, Iter-RetGen produces an answer in each iteration, with each iteration consisting of a single round of retrieval and generation. We exploit the synergy between retrieval and generation with multiple iterations. The number of iterations is a hyper parameter, and usually, 2 or 3 iterations suffice to give the most performance boost. As can be seen in Table 3, Iter-RetGen using 2 iterations already achieves significantly higher or competitive accuracy while causing fewer overheads of retrieval and generation than ReAct and Self-Ask.
>
> > The improvement of retrieval is mostly due to the distillation process, not by the inclusion of y1. However, the method to distill the reranker to retrieval has been proposed in Re2G
>
> We used the off-the-shelf Contriever-MSMARCO for all experiments except those in Table 4. In Table 2, the improvements of Iter-RetGen over baselines come from synergizing retrieval and generation by iterating retrieval-augmented generation and generation-augmented retrieval.
>
> We also shown that generation also benefits retrieval adaptation. Apart from distilling knowledge from a re-ranker, we also include model generation $y_1$ to the optimization process, so that we can (1) retrieve more relevant documents for the retriever to train on and (2) help the retriever better address the semantic gaps between a question and its supporting knowledge by using $y_1$ as an additional input to the teacher model (i.e., the re-ranker).
>
> As shown in Table 4, `distillation with $y_1$` consistently outperforms `distillation without $y_1$` on both HotPotQA and Feverous, with significant improvement (3.4\% absolute gain) on Feverous.
>
> > Can you provide some insights on how the performance of ITER-GEn compared to Re2G? That is to say, we finetune the generation based on the retrieval outputs instead of using in-context learning?
>
> As different tasks have different answer spaces (e.g., short-form qa vs. boolean qa), we used in-context learning mainly to ensure the model generates task-specific answers. Finetuning the generation model is definitely an alternative way to shape the output space, and potentially improves utilization of retrieved knowledge.
>
> Iter-RetGen is based on the observation that a model-generated answer can bridge the semantic gaps between a question and its supporting knowledge, which we believe is applicable to generation systems including Re2G, i.e., to retrieve more relevant knowledge based on generated answers, and then generate better answers based on updated knowledge.
>
> > Missing Reference to the Re2G paper
>
> Thanks for the pointer! We will cite this paper.

---

### Official Review · Reviewer_vbix · 2023-08-04

**Soundness:** 3

**Excitement:**

3: Ambivalent: It has merits (e.g., it reports state-of-the-art results, the idea is nice), but there are key weaknesses (e.g., it describes incremental work), and it can significantly benefit from another round of revision. However, I won't object to accepting it if my co-reviewers champion it.

**Paper Topic And Main Contributions:**

This paper propose ITER-RETGEN to synergize retrieval and generation in an iterative manner to enhance large language models. And the authors also adapts retrieval with model generations. Empirical results show that ITER-RETGEN performs better on three different tasks than other retrieval-augmented methods. In addition, the authors propose a reliable evaluation metric, which uses text-davinci-003 to evaluate  the correctness rather than the exact match.

**Questions For The Authors:**

1. What is the length of each candidate Wikipedia chunk?
2. Table 1 is not necessary and can be placed in the appendix.
3. Why ITER-RETGEN is worse than self-ask on MuSiQue? Why does CoT perform almost best on Bamboogle?

**Reasons To Accept:**

1. This paper proposes a simple and effective method to enhance retrieval-augmented LLMs through retrieval-generation synergy.
2. This paper introduce a reliable evaluation metric, which is more rubust than exact match.

**Reasons To Reject:**

1. In lines 65-70, the authors mention the limitations of previous work, but ITER-RETGEN seems to have similar problems.


**Reproducibility:**

4: Could mostly reproduce the results, but there may be some variation because of sample variance or minor variations in their interpretation of the protocol or method.

**Reviewer Confidence:**

4: Quite sure. I tried to check the important points carefully. It's unlikely, though conceivable, that I missed something that should affect my ratings.

---

> ### Author Rebuttal · Authors · 2023-08-29
>
> Thanks for your time, insightful comments, and questions. We provided our responses below.
>
> > In lines 65-70, the authors mention the limitations of previous work, but ITER-RETGEN seems to have similar problems.
>
> The limitations we mentioned apply to the structured workflows adopted by previous work like ReAct and Self-Ask.
>
> * Limitation 1: `they fail to process all retrieved knowledge as a whole during the generation process`
>
>   * Methods like ReAct and Self-Ask interleave retrieval and generating intermediate reasoning steps, until the model generates the final answer. Each intermediate reasoning step is conditioned on knowledge retrieved before, with no awareness of knowledge retrieved afterwards.
>   * **How does Iter-RetGen alleviate this limitation?** In each iteration, Iter-RetGen retrieves knowledge based on (1) the question and (2) the previous model output which shows what may be needed to answer the question. Unlike ReAct and Self-Ask conditioning each reasoning step on partial knowledge, Iter-RetGen generates reasoning steps and the final answer based on all knowledge retrieved in an iteration. Notably, knowledge retrieved in an iteration is not complementary to that in the previous iteration, but tends to be more comprehensive and relevant. In our preliminary experiments, we have tried conditioning reasoning steps and the final answer on a merged version of knowledge from different iterations, but found no noticeable improvements.
>
> * Limitation 2: `they invoke retrieval multiple times, and may frequently change the prompts by updating newly retrieved knowledge, thus increasing the overheads of both retrieval and generation`
>
>   * To finalize a single answer, methods like ReAct and Self-Ask have to invoke multiple rounds of retrieval and generation to gather a comprehensive set of knowledge, with different rounds focusing on complementary aspects and updating the prompt accordingly.
>   * **How does Iter-RetGen alleviate this limitation?** Unlike ReAct and Self-Ask invoking multiple rounds of retrieval and generation to finalize a single answer, Iter-RetGen produces an answer in each iteration, with each iteration consisting of a single round of retrieval and generation. We exploit the synergy between retrieval and generation with multiple iterations. The number of iterations is a hyper parameter, and usually, 2 or 3 iterations suffice to give the most performance boost. As can be seen in Table 3, Iter-RetGen using 2 iterations already achieves significantly higher or competitive accuracy while causing fewer overheads of retrieval and generation than ReAct and Self-Ask.
>
> >  What is the length of each candidate Wikipedia chunk?
>
> We used preprocessed wikipedia chunks from previous works. Chunks from the 2017, 2018, 2021 Wikipedia dumps are of size no more than 100, 100, 200 words, respectively.
>
> > Table 1 is not necessary and can be placed in the appendix.
>
> Thanks for the suggestion! We will place it in the appendix.
>
> > Why ITER-RETGEN is worse than self-ask on MuSiQue?
>
> Iter-RetGen is competitive with Self-Ask on MuSiQue, but with a slightly lower accuracy. We manually investigated 10 random MuSiQue questions where Iter-RetGen (T=2) fails but Self-Ask gives correct answers. For Iter-RetGen, 7 / 10 of error cases are related with retrieval, on 2 of which reasoning in the first iteration is totally correct but the retriever retrieves distractive facts in the second iteration, while on the other cases, reasoning in the first iteration is partially correct, but the retriever fails to retrieve the missing pieces in the second iteration.
>
> > Why does CoT perform almost best on Bamboogle?
>
> Text-davinci-003 is knowledgeable, and can achieve almost the best accuracy using its own parametric knowledge (i.e., using CoT without retrieval) on Bamboogle. Iter-RetGen can achieve a slightly higher accuracy while both ReAct and Self-Ask perform significantly worse.
>
> On Bamboogle, all retrieval-augmented methods evaluated in this paper benefit Text-davinci-003 no more often than introducing noisy or even distractive facts. We manually investigated 10 random Bamboogle questions where both CoT without retrieval and Iter-RetGen give wrong answers. For 60% of them, we failed to retrieve relevant information using our retrieval system even with clearly-stated queries:
>
> * The dense retriever fails to capture fine-grained information needs, e.g., for the query *When did George Washington leave office?*, the dense retriever fails to retrieve `Washington was inaugurated as the first president of the United States, and ended on March 4, 1797`, but returns `Washington decided early in 1796 that he would retire... departure from office after two terms`.
> * The retrieval corpus is not comprehensive, e.g., the 2021 Wikipedia dump seems to contain no information about *How much protein is in one boiled egg yolk?*

---

### Meta-Review · Area_Chair_YVzD · 2023-09-15

**Recommendation:** 2

**Metareview:**

This paper proposes a new method to synergize retrieval and generation iteratively by using a response from a previous iteration at a retrieval step to refine the output. Evaluation on 4 multi-hope QA datasets, 1 fact verification and 1 commonsense task show consistent improvements.

Reviewers agreed that the method is simple yet effective (vbix, yjLF, ij2Q), and acknowledged that experiments are comprehensive, along with a new, more reliable metric proposed (vbix, yjLF, ij2Q) and insightful analysis (yjLF).

However, two reviewers pointed out limited novelty (yjLF, ij2Q). Also, reviewers mentioned how the method solves the issues with previous method mentioned in the paper—failing to process all retrieved knowledge and needing to use retrieval multiple times—is unclear (vbix, yjLF). Authors are encouraged to revise the paper to address this concern in the next version of the paper.

---

### Decision · Program_Chairs · 2023-10-07

**Decision:**

Accept-Findings

**Comment:**

This paper proposes a new method to synergize retrieval and generation iteratively by using a response from a previous iteration at a retrieval step to refine the output. Evaluation on 4 multi-hope QA datasets, 1 fact verification and 1 commonsense task show consistent improvements.

Reviewers agreed that the method is simple yet effective (vbix, yjLF, ij2Q), and acknowledged that experiments are comprehensive, along with a new, more reliable metric proposed (vbix, yjLF, ij2Q) and insightful analysis (yjLF).

However, two reviewers pointed out limited novelty (yjLF, ij2Q). Also, reviewers mentioned how the method solves the issues with previous method mentioned in the paper—failing to process all retrieved knowledge and needing to use retrieval multiple times—is unclear (vbix, yjLF). Authors are encouraged to revise the paper to address this concern in the next version of the paper.